# Morin-VitaminE-β-CyclodextrinInclusionComplexLoadedChitosanNanoparticles (M-Vit.E-CD-CSNPs) Ameliorate Arsenic-Induced Hepatotoxicityina Murine Model

**DOI:** 10.3390/molecules27185819

**Published:** 2022-09-08

**Authors:** Sanchaita Mondal, Sujata Das, Pradip Kumar Mahapatra, Krishna Das Saha

**Affiliations:** 1Cancer Biology and Inflammatory Disorder Division, CSIR-Indian Institute of Chemical Biology, 4 Raja S.C. Mullick Road, Kolkata 700032, West Bengal, India; 2Department of Chemistry, Jadavpur University, 188 Raja S.C. Mullick Road, Kolkata 700032, West Bengal, India

**Keywords:** β-cyclodextrin, Morin, vitamin E, chitosan, inflammation, oxidative stress, apoptosis, nanoformulation, hepatoprotection

## Abstract

The special features of cyclodextrins (CDs), hydrophilic outer surfaces and hydrophobic inner surfaces, allow for development of inclusion complexes. The two bioactive strong antioxidant hepatoprotective compounds, Morin and vitamin E, are water insoluble. The present study aimed to prepare Morin-vitamin E-β-cyclodextrin inclusion complex loaded chitosan nanoparticles (M-Vit.E-CD-CS NPs) and to examine their hepatoprotective efficacy against arsenic-induced toxicity in a murine model. The NPs were characterized by FTIR, DLS, NMR, DSC, XRD, AFM, and a TEM study. The NPs were spherical in shape, 178 ± 1.5 nm in size with a polydispersity index (PDI) value of 0.18 and a zeta potential value of −22.4 ± 0.31 mV, with >50% encapsulation and drug loading efficacy. Mice were exposed to arsenic via drinking water, followed by treatment without or with the NPs on every alternate day up to 30 days by oral gavaging. Administration of NPs inhibited the arsenic-induced elevation of liver function markers, inflammatory and proapoptotic factors, reactive oxygen species (ROS) production, alteration in the level of blood parameters and antioxidant factors, and liver damage, which was measured by different biochemical assays, ELISA, Western blot, and histological study. Organ distribution of nanoparticles was measured by HPLC. M-Vit.E-CD-CS NPs showing potent hepatoprotective activity may be therapeutically beneficial.

## 1. Introduction

Cyclodextrins (CDs) are natural, cyclic oligosaccharides that are non-toxic in nature. They contain a hydrophilic outer surface as the hydroxyl groups are faced to the outward direction and a lipophilic central cavity as the hydrogen and glycosidic oxygen bonds are facing inside the core. Therefore, cyclodextrin acts as a host, and it can form an inclusion complex interacting through soft bonds with a guest compound in the hydrophobic region, without affecting the host framework structure [1]. Among three types of CDs, *β*-CD is commonly used because the size of the internal pocket of β-CD is appropriate for the guest’s molecule with molecular weights 200–800 g/mol. However, β-CD is able to interact with different types of molecules in higher molecular weights by a specific moiety [2]. Several studies have suggested that β-CD can enhance the drug loading efficiency of nanoparticles and decelerate the release of drugs.

Chitosan (CS), as a biocompatible and biodegradable polymer, has been heavily used in nanoparticle formation. Chitosan has drug release ability in a controlled and a sustained way and good solubility in an aqueous acidic solution [3]. Several researchers have reported the application of CS nanoparticles for the delivery of hydrophobic drugs [4,5] along with hydrophilic protein [6,7].

Morin hydrate (2′,3,4′,5,7-pentahydroxyflavone), a yellowish bioflavonoid, mainly obtained from the fruits, stem, and leaves of Moraceae family member plants [8] prevents a broad spectrum of disease pathologies, including liver toxicity, diabetes, ischemia, cardiovascular anomalies, cancer, neurotoxicity, and renal complications [9]. Research has shown that Morin administration does not show any harmful outcomes [10]. Besides, it is inexpensive and readily obtainable [11]. Morin exhibits a protective effect due to its strong antioxidant efficacy and its distinct architecture helps to bind it with nucleic acids, enzymes, and proteins [12]. Vitamin E is a naturally available, fat-soluble antioxidant [13]. Its hepatoprotective activity results from its scavenging effect on ROS [14].

In the arsenic-induced liver injury serum level of asparatate amino transferase (AST), alanine amino transferase (ALT) and alkaline phosphatase (ALP) become uplifted [15]. Excess Reactive Oxygen Species (ROS) are generated due to arsenic intoxication in addition to quick attenuation of antioxidant enzymes, such as catalase and superoxide dismutase (SOD) [16]. The leading ROS-affected pathways following arsenic exposure include Nrf2-antioxidant response element (ARE) signaling pathways [17]. Nrf2, a prime transcription factor in antioxidant system, remains bound to kelch-like epichlorohydrin-associated protein1 (Keap1) in the cytoplasm [18]. Under oxidative stress or in response to antioxidant agents, Nrf2 is dissociated from Keap1 and translocated to the nucleus, where it binds to the ARE motif and activates the production of numerous antioxidant enzymes and detoxification enzymes, e.g., superoxide dismutase (SOD), Catalase (CAT), glutathione (GSH), glutathione-S-transferase (GST), GSH peroxidase (GPx), NADPH quinone oxidoreductase 1(NQO1), heme oxygenase-1 (HO-1), etc., to neutralize the ROS [19,20,21]. Excessive levels of ROS induce apoptotic signaling pathways. Therefore, the inhibition of apoptosis in liver tissue is one of the indicators to explore the protective efficacy of drugs.

NF-κB is responsible for transcriptional induction of pro-inflammatory cytokines, such as IL-1, IL-6, IL-12, TNF-α, chemokines, and additional inflammatory mediators [22]. NF-κB also has a role in controlling the activation of NLRP3 inflammasomes. So, deregulated NF-κB activation is the sign of chronic inflammatory diseases. The elevated level of NF-κB and NLRP3 confirm the inflammatory responses in arsenic-induced liver injury.

So, in the present study we have synthesized and characterized MOR-Vitamin E-β-CD inclusion complex loaded chitosan nanoparticles and evaluated its hepatoprotective role through measurement of inhibition of excessive ROS production, inflammation and apoptotic responses, and improvement of antioxidant factors.

## 2. Results

### 2.1. Characterization of M-Vit.E-CD-CS NPs

MOR-Vit.E-β-CD inclusion complex loaded chitosan NPs (M-Vit.E-CD-CS NPs) were prepared by two steps and characterized. Fourier Transform IR (FTIR) exhibits the compatibility between MOR-Vit.E-β-CD inclusion complex encapsulated CS nanoparticle (Figure 1A). The FT-IR spectrum covers the range from 4000 cm^−1^ to 400 cm^−1^. The notable peaks attributed for MOR, vitamin E, β-CD, chitosan, and M-Vit.E-CD-CS NPs validate the existence of their distinct functional groups. The strong peak at 3400 cm^−1^ signifies O-H (stretching), an absorption band at ~1600 cm^−1^ indicates the existence of alkene (C=C) groups, and the sharp peak observed at ~1180 cm^−1^ denotes C-OH (stretching) in MOR, vitamin E, β-CD, and chitosan. The M-Vit.E-CD-CS NPs show a band at 3400 cm^−1^, which indicate that the hydroxyl groups of MOR, vitamin E, β-CD, and CS are preserved. Additional short humps at 1640 cm^−1^ and 1150 cm^−1^ imply the protection of the alkene (C=C) and the ester (C=O) groups and the C-OH bond of MOR, vitamin E, β-CD, and chitosan in the M-Vit.E-CD-CS NPs. The DLS data shows that the size of M-Vit.E-CD-CS NPs is 178 ± 1.5 nm, as shown in Figure 1B, with a polydispersity index (PDI) value of 0.18. The zeta potential value of M-Vit.E-CD-CS NPs is −22.4 ± 0.31 mV, and it has a propensity to balance the suspension of NPs. Atomic force microscopy (AFM) and Transmission Electron Microscopy (TEM) show that the outward topology of M-Vit.E-CD-CS NPs is spherical, and they are systematically divided, without conglomeration. The size of M-Vit.E-CD-CS NPs ranges within 100–200 nm (Figure 1C,D).

The encapsulation efficacy of M-Vit.E-CD-CS NPs for MOR is 78 ± 2.5% and for vitamin E 73 ± 1.8%. Earlier reports suggest that the NPs with a spherical shape show a high level of drug encapsulation efficacy. The drug loading efficacy of M-Vit.E-CD-CS NPs is 58 ± 3.1% for MOR and 42 ± 1.5% for vitamin E (Figure 1E). The in vitro drug release kinetics of M-Vit.E-CD-CS NPs are shown in Figure 1F. A total of 60% MOR and 52% vitamin E are released from the nanoparticles in 24 h. A prolonged release of MOR and vitamin E were monitored up to 4 days, but the highest release was detected at 72 h, which was 82.5% for MOR and 71.8% for vitamin E.

^1^HNMR was recorded for MOR, Vit.E, β-CD, chitosan, and M-Vit.E-CD-CS NPs (Figure 2A). The results showed that in nanoparticles there is no significant peak of MOR and vitamin E. Only the peaks of chitosan and cyclodextrin were observed, which again proved the successful encapsulation of the drug in the cyclodextrin cavity.

TGA data showed the % weight loss of NPs from 235 °C (Figure 2B,C). In the case of NPs, the DSC thermogram showed an endothermic peak at 84.33 °C (Figure 2D), which is quite similar to the Tg values of chitosan nanoparticles reported in the literature [23]. That indicated the incorporation of the M-Vit.E-CD inclusion complex in the polymer matrix of chitosan. XRD spectra were recorded for the nanoparticles, and they revealed the loss of crystalline character of the NPs as no sharp peak was observed (Figure 2E). This indicated that due to formation of the nanoparticles, the crystallinity of the used compounds was lost. The XRD value was consistent with the results obtained from DSC studies.

### 2.2. Effect of MOR, Vitamin E and M-Vit.E-CD-CS NPs on Arsenic Treated HepG2 Cells

The mortality rate of cells due to arsenic exposure was analyzed by MTT assay. The cells were treated with different concentrations of arsenic (5, 10, 20, 30, 40 µg/mL) up to 24 h and then the cell death percentage was evaluated (Appendix A). To further investigate the role of MOR, vitamin E, and M-Vit.E-CD-CS NPs in As (III)–induced liver cell dysfunction, the cell survival percentage was detected in the HepG2 cell line. The result showed that there was a decrease in the cell survivability percentage with the treatment of arsenic, which increased with the treatment of MOR, vitamin E, and M-Vit.E-CD-CS NPs (Figure 3A).

ROS generation triggers cell death. So, ROS level was fluorometrically studied in arsenic cotreated with MOR, vitamin E, and M-Vit.E-CD-CS NPs in HepG2 cells. An increased ROS level was observed with the treatment of arsenic, which was altered following the treatment of MOR, vitamin E, and M-Vit.E-CD-CS NPs (Figure 3B). This result also indicated that the effect of M-Vit.E-CD-CS NPs was higher than MOR and vitamin E.

DNA fragmentation is the indication of cell death. So, we investigated the DNA fragmentation level induced by arsenic in HepG2 cells and the effect of MOR, vitamin E, and M-Vit.E-CD-CS NPs on it. The level of fragmented DNA increased with arsenic treatment and decreased with the treatment of MOR, vitamin E, and M-Vit.E-CD-CS NPs in HepG2 (Figure 3C). A similar rise in DNA fragmentation similar to the ROS level was observed, and the effect of M-Vit.E-CD-CS NPs was comparable the ROS-inhibitor NAC. In addition, the only noticeable part of this study was the impact of the M-Vit.E-CD-CS NPs on the hepG2 cell line; they did not affect the survivability of the HepG2 cell line.

### 2.3. Effect of MOR, Vitamin E, and M-Vit.E-CD-CS NPs on Liver Function Markers, Changing Body Weight, Hematological Parameters, Kidney Function Markers, and Lipid Profiles

A non-poisonous dose of MOR was 200 mg/kg as reported earlier [24]. Different doses of vitamin E and M-Vit.E-CD-CS NPs were orally administered in mice, and the effect of them on the serum level of AST and of ALT were measured. On every alternate day up to 50 mg/kg of vitamin E and 20 mg/kg of M-Vit.E-CD-CS NPs given orally for 30 days did not elevate the serum level of ALT and AST (Appendix A). Although, the serum level of ALT and AST uplifted following the treatment of 100 mg/kg body weight of vitamin E and 40 mg/kg body weight of M-Vit.E-CD-CS NPs. So, the effect of MOR, vitamin E, and M-Vit.E-CD-CS NPs against arsenic intoxification was studied using the highest doses 200 mg/kg, 50 mg/kg, and 20 mg/kg, respectively.

We have taken a 40 mg/L dose of arsenic as per our previous report [25]. The serum levels of AST and ALT elevated gradually in arsenic-induced mice (40 mg/L). Treatment of MOR (50, 100, and 200 mg/kg bwt), vitamin E (10, 25, and 50 mg/kg bwt), and M-Vit.E-CD-CS NPs (5, 10, and 20 mg/kg bwt) every other day in the time of arsenic exposure dose-dependently reduced the levels of AST and ALT, as shown in Figure 4A,B. The optimum effect was observed with 20 mg/kg M-Vit.E-CD-CS NPs, 200 mg/kg bwt of MOR, and 50 mg/kg bwt of vitamin E. Thus, the potency of M-Vit.E-CD-CS NPs was nearly 10 times higher than MOR and 2.5 times higher than vitamin E.

During arsenic exposure, the body weight of mice decreased along with the RBC level; platelet (PLT), hemoglobin (Hb), and HDL (high-density lipoprotein) also reduced; and WBC (white blood cells), LDH (lactate dehydrogenase), kidney function markers, e.g., urea, uric acid, and creatinine, and lipid markers, e.g., cholesterol, triglycerides, and LDL (low-density lipoprotein) became uplifted. An elevated level of LDH suggests tissue damage. Treatment of mice with 20 mg/kg bwt M-Vit.E-CD-CS NPs, 200 mg/kg bwt of MOR, and 50 mg/kg bwt of vitamin E inverted the arsenic promoted transformation in body weight, level of kidney function markers, and blood parameters (Table 1). In addition, following the treatment with 20 mg/kg bwt M-Vit.E-CD-CS NPs, 200 mg/kg bwt of MOR, and 50 mg/kg bwt of vitamin E increased the serum level of LDL (low-density lipoprotein), TG (Triglyceride), and TC (Total cholesterol), the moderate level of HDL was remarkably changed (Table 1).

### 2.4. Effect of MOR, Vitamin E, and M-Vit.E-CD-CS NPs on Arsenic Accumulation in Various Organs

The effect of vitamin E, MOR, and M-Vit.E-CD-CS NPs treatment on arsenic accumulation in liver, brain, lung, heart, skin, and kidney are represented in tabular form. Arsenic exposure results in a remarkable rise in arsenic levels in these organs. It also suggests that the arsenic accumulation is higher in the liver than in the other organs. Treatment of M-Vit.E-CD-CS NPs, vitamin E, and MOR reduced the arsenic accumulation in these organs (Table 2).

### 2.5. M-Vit.E-CD-CS NPs Suppress Oxidative Stress in Mice Liver

Administration of vitamin E, MOR, and M-Vit.E-CD-CS NPs significantly hindered ROS generation and increased the level of MDA, which is induced by arsenic exposure (Figure 4C,D).

### 2.6. Effect of M-Vit.E-CD-CS NPs on Arsenic-Induced Elevation of Antioxidant Factors

In the period of arsenic exposure, the level of antioxidant factors and antioxidant enzymes are reduced. So, we investigated the effect of vitamin E, MOR, and M-Vit.E-CD-CS NPs on an arsenic-induced decrease in antioxidant factors, such as SOD, catalase, GSH. SOD was up-regulated by vitamin E (50 mg/kg), MOR (200 mg/kg), and M-Vit.E-CD-CS NPs (20 mg/kg) in arsenic-exposed mice (Figure 4A). The lowered levels of catalase and GSH in arsenic-treated mice receiving M-Vit.E-CD-CS NPs increased (Figure 5B,C). Due to arsenic intoxification, the level of nuclear Nrf2 and GPx decreased, and they further increased, following the treatment of M-Vit.E-CD-CS NPs (Figure 5D). Additionally, the study showed that cytosolic Nrf2 was notably elevated in the arsenic-exposed mice, which was lowered, following the treatment of M-Vit.E-CD-CS NPs (Figure 5D). Identical results were obtained in the amount of HO-1 and NQO1 (Figure 5D). The effect of 20 mg/kg bwt M-Vit.E-CD-CS NPs was comparable with 200 mg/kg of MOR and 50 mg/kg of vitamin E. This indicated that M-Vit.E-CD-CS NPs have a powerful antioxidant activity against arsenic-induced liver damage.

### 2.7. M-Vit.E-CD-CS NPs Inhibit Apoptosis in Liver Tissue Induced by Arsenic

Arsenic-induced inflation of DNA fragmentation, active caspase-3/caspase-9, and cytosolic cytochrome C were conquered by M-Vit.E-CD-CS NPs in the liver tissue lysate (Figure 6A,B). The effect of M-Vit.E-CD-CS NPs on DNA fragmentation was compared with the ROS-inhibitor NAC, which showed that the effect of M-Vit.E-CD-CS NPs was nearly the same as NAC (Figure 6C). In the treated group of mice by arsenic + MOR, arsenic + vitamin E, and arsenic + M-Vit.E-CD-CS NPs, the level of Bax, Bad, p53, Apaf-1, and PUMA were found to be lowered in comparison to their elevated level in the arsenic group. Arsenic exposure suppressed the level of anti-apoptotic protein Bcl-2, which was enhanced by the following treatment of MOR, vitamin E, and M-Vit.E-CD-CS NPs (Figure 6D).

### 2.8. M-Vit.E-CD-CS NPs Suppressed Arsenic-Induced Inflammatory Responses

The formation of pro-inflammatory mediators was enhanced due to the separation of IkBα from NF-kB in cytosol, accelerating its shifting into the nucleus [26]. In the time of arsenic exposure, NF-kBp65 is extremely triggered with the release of a prominent level of inflammatory cytokines. The treatment of MOR, vitamin E, and M-Vit.E-CD-CS NPs reduced the amount of these cytokines, such as IL-1β, IL-6, and TNF-α, increased by arsenic (Figure 7A). Arsenic exposure activated NF-kBp65, which was further suppressed, following the treatment of MOR, vitamin E, and M-Vit.E-CD-CS NPs (Figure 7B). The activation of NLRP3 inflammasome occurred due to chronic arsenic exposure and the release of pro-inflammatory factors, e.g., Caspase-1 and IL-18, cause pyroptosis. The increased level of NLRP3 inflammasome caused by arsenic and the amount of Caspase-1 and IL-18 in liver tissue was attenuated by M-Vit.E-CD-CS NPs treatment (Figure 7B).

### 2.9. M-Vit.E-CD-CS NPs Ameliorate Liver Tissue Damage Induced by Arsenic

The histopathological changes were analyzed by using H&E staining in the liver tissue section. The control liver tissue showed the cords of the normal hepatocytes in which sinusoid lines were looking normal by Kupffer cells, whereas diffused Kupffer cells, vascular changes, and infiltration of inflammatory cells showed arsenic exposure, indicating critical liver damage (Figure 8A). The repair of liver tissue took place in arsenic-intoxicated mice injected with M-Vit.E-CD-CS NPs as compared with the free MOR and vitamin E treated mice (Figure 8A).

### 2.10. Tissue Distribution Study of M-Vit.E-CD-CS NPs in Various Organs

A single dose of M-Vit.E-CD-CS NPs orally given to the mice for several time points and the concentration of Morin and vitamin E in liver, spleen, lungs, kidneys, and serum were recorded by HPLC. Figure 8A showed the different concentration of Morin from M-Vit.E-CD-CS NPs in different organs. At the time point of 2 h, the highest concentration of Morin was present in the liver, and it was reduced slowly with time, and it was the lowest at 72 h. The sequence of the amount of Morin and vitamin E in various organs was liver> kidney ≥ spleen > lung > serum. With the period of time, the concentration of Morin and vitamin E in different organs steadily decreased, indicating nominal or no deposition in the tissues and a gradual removal of the components (Figure 8B).

## 3. Discussion

CDs are widely used in pharmaceuticals, drug delivery systems, cosmetics, and the food and chemical industries. They can be found in commercially available medications, including tablets, eye drops, and ointments [27]. Their successful use in inclusion complexes with bioactive compounds has led to extensive investigations in several different application areas to try to overcome the limitations of certain substances. In the food industry, CDs are used in the reduction of cholesterol in food, as dietary fibers, for controlling body weight and blood lipid profile, and as prebiotics, which enhance the intestinal microflora [28]. Although CDs are either non or only partly digestible by the enzymes of the human gastrointestinal (GI) tract and fermented by the gut microflora, they produce negligible cytotoxicity (mainly β-CD) in the body.

β-CD and chitosan are both used as nanodrug delivery systems. However, there are previous reports that suggest CD-g-CS nanoparticles showed a better stability in comparison to chitosan nanoparticles [29]. CDs can regulate various properties, affecting the performance and the therapeutic uses of drugs. Besides enhancing the hydrophilicity and the rate of dissolution of poorly water-soluble drugs, CDs can also reduce harmful reactions, e.g., gastrointestinal or ocular irritation and other side effects [27]. They increase absorbency through the biological membranes, reduce vaporization, and stabilize flavors, which enhances tastiness. Several studies have suggested that β-CD can enhance the drug loading efficacy of nanoparticles and decelerate the release of drugs. CD inclusion complexes loaded chitosan nanoparticles can result in mucoadhesive delivery systems, having the combined effects of inclusion, bioavailability enhancement, and specific mucosal targeting.

Morin, a poorly water soluble flavonoid, and Vitamin E, a fat soluble antioxidant nutrient were chosen for testing the ability to load hydrophobic drugs of the best NPs formulations in this study. The synthesized NPs were characterized by using various techniques. The results show that Morin and Vitamin E were entrapped in the cavity of cyclodextrins. The inclusion complexes in which a guest molecule was surrounded by a hydrophobic environment of CDs cavity is ideal for delivering a low solubility drug. Inclusion complexes have also been studied in order to significantly improve the hepatoprotective of Morin and Vitamin E. Morin was sustainably released, but a higher antioxidant and hepatoprotective activity was obtained upon inclusion complexed with β cyclodextrin. Results indicate that CDs could facilitate the association of a complexed drug into the NPs. Inclusion complexes formed with a host–guest molecule may exhibit improved chemical or biological properties compared to the host molecule alone.

Chitosan NPs also exhibit significant hepatoprotective effect in the handling of liver injury induced by alcohol, CCl4, acetaminophen, diethylnitrosamine (DEN), and concanavalin A [30,31,32,33]. Chitosan-graft-β-cyclodextrin nanoparticles were reported as a good carrier for controlled drug release [29].

Both Morin and vitamin E have been reported to have a hepatoprotective role in various disease models [14,34].

However, their hydrophobic nature is the reason for poor bioavailabilty. To counter these drawbacks, we have synthesized β-CD-MOR-Vit.E inclusion complex loaded chitosan nanoparticles, called M-Vit.E-CD-CS NPs, which have a prominent aqueous solubility and therefore an increased bioavailability. The efficacy of M-Vit.E-CD-CS NPs is much higher than that of MOR or vitamin E. The conservation of the significant FT-IR peak of the free compounds confirmed the formation of the nanoparticles. The average size of the nanoparticles is 178 ± 1.5 nm. The TEM and AFM images confirmed the spherical shape of the nanoparticles.

The size of nanoparticle plays a crucial role, and it is key to achieving therapeutic efficacy, depending on the physiological parameters, such as blood circulation half-life, molecular targeting, and cellular uptake. Nanoparticles that fall between 5–200 nm tend to have a prolonged blood circulation [35]. Tissue distribution studies have revealed that M-Vit.E-CD-CS NPs accumulation increased in the liver at 2 h and gradually decreased up to 72 h. This result suggested the slow elimination of nanoparticles from blood circulation. The shape of our synthesized nanoparticles were spherical, and it also justified the previous report by Wang et al., which suggested that intake of the spherical MSNs by liver increased constantly to the 72-h time point.

Cotreatment of MOR, Vitamin E, or M-Vit.E-CD-CS NPs with arsenic in mice reduced liver function markers, ROS level, pro-apoptotic and inflammatory factors with elevation of antioxidant parameters, and improvement of liver histopathology. Interestingly, the effect of M-Vit.E-CD-CS NPs treatment showed better effects than free Morin or vitamin E in all cases. All these beneficial outcomes can be associated to the therapeutic efficacy of M-Vit.E-CD-CS NPs.

## 4. Materials and Methods

### 4.1. Chemicals

Morin hydrate (purity ≥ 99.0%), chitosan (low MW, extrapure), dichloromethane (DCM), β-Cyclodextrin (β-CD), vitamin E (α-Tocopherol), and Sodium TPP were purchased from Sisco Research Laboratories (Gurugram, India), and all primary and secondary antibodies were purchased from Sigma (St. Louis, MO, USA). Assays kits for the detection of serum ALT, AST, AP, SOD, catalase, GPx, HDL, LDL, triglyceride (TG), and total cholesterol (TC) were bought from ARKRAY Healthcare Pvt. Ltd. (Surat, India). IL-1β, IL-6, TGF-β, and TNF-α were measured by ELISA kits from R&D System (Minneapolis, MN, USA). All other chemicals were available commercially and of a high degree of purity. Dulbecco’s modified Eagle’s medium (DMEM), fetal bovine serum (FBS), penicillin, streptomycin, neomycin (PSN) antibiotic, ethylenediaminetetraaceticacid (EDTA), and trypsin were bought from Gibco BRL (Grand Island, NY, USA). 3-(4, 5-Dimethylthiazol-2-yl)-2,5-diphenyltetrazolium bromide (45,989, MTT-CAS 298-93-1-Calbiochem) and DMSO were bought from Merck-Millipore (Burlington, MA, USA). Tissue culture plastic wares were bought from Genetix Biotech Asia Pvt. Ltd. (Delhi, India). Zinc acetate was purchased from Sigma-Aldrich. A HepG2 cell line was obtained from the National Centre for Cell Science (NCCS), Pune, India.

### 4.2. Preparation of MOR-Vit.E-β-CD Inclusion Complex

A particular amount of β-CD (15 mg/mL) was dissolved in deionized water in a glass tube containing a magnetic bar. To this solution, equivalent amounts of MOR and vitamin E (1:1 ratio) in 70 μL acetone were prepared and stirred for 16 h [36]. Stirring was allowed to evaporate acetone without a cap. The solution was stirred overnight until a clear solution was acquired. The solution was then centrifuged at 4000 rpm for 15 min, and a supernatant containing highly water soluble MOR-Vit.E-β-CD inclusion complexes was dried in a freeze dryer (DFU-1200, Tokyo Rikakikai Co., Ltd., Tokyo, Japan). The dried MOR-Vit.E-β-CD inclusion complexes were stored at 4 °C until further use.

### 4.3. Preparation of MOR-Vit.E-β-CD Inclusion Complex Loaded Chitosan NPs

MOR-Vit.E-β-CD inclusion complex loaded chitosan NPs (M-Vit.E-CD-CS NPs) were synthesized by using the ionic gelation method. Briefly, a desired amount of chitosan dissolved in acetic acid (1%) was mixed with a certain amount of MOR-Vit.E-β-CD inclusion complexes solution at room temperature. After 5 min of stirring, the TPP solution was added dropwise to the mixture to form nanoparticles. For the preparation of nanoparticles with the highest encapsulation efficacy, chitosan, MOR-Vit.E-β-CD inclusion complexes, and TPP were used in a 3:5:5 ratio, respectively, according to Bardania et al. [37]. The solution of the prepared nanoparticles was centrifuged at 10,000 rpm for 10 min. To obtain the respective NP powder, M-Vit.E-CD-CS NPs pellets were lyophilized for 3 days and preserved at −20 °C.

### 4.4. Determination of MOR and Vitamin E Encapsulation Efficiency and Drug-Loading

NPs (5 mg) were soaked in 5 mL of phosphate buffer for 30 min. The solution was centrifuged at 4000 rpm at 4 °C for 40 min, and the precipitate was washed twice with fresh solvent to remove the unconjugated drug. Using a UV spectrophotometer (JASCO V-730, Spectrophotometer, Tokyo, Japan) at λ_max_ values of 270 and 395 nm, the clear supernatant solution was analyzed for unencapsulated Morin and vitamin E. Standard curves of Morin and vitamin E were obtained by plotting the concentration against the absorbances from 10 to 50 mg/mL.

The percentages of drug loading and entrapment efficiency were calculated by using the following formula:Encapsulation efficiency (%) = (The total amount of drug released from the lyophilized M-Vit.E-CD-CSNPs/Amount of drug initially taken to synthesize the M-Vit.E-CD-CS NPs) × 100.
Drug loading (%) = (Amount of drug found in the lyophilized M-Vit.E-CD-CS NPs/Amount of lyophilized M-Vit.E-CD-CS NPs) × 100.

### 4.5. Particle Size Measurement

A Zetasizer 3000 HSA (Malvern Instruments, Malvern, UK) was used to measure the particle size and the distribution. Using 12 mm cells at 90 degrees and a temperature of 25 °C, differential light scattering (DLS) was used to determine the mean NP diameter. Before the tests, the NPs were diluted with double-distilled water, and 500 μL was loaded into the cuvette for DLS and polydispersity index readings.

### 4.6. Fourier Transform Infrared Spectroscopy

To identify the various functional groups present in Morin, chitosan, β-cyclodextrin, vitamin E, and M-Vit.E-CD-CS NPs, an FT-IR was performed. All the compound and synthesized NPs, which were centrifuged and lyophilized earlier; and the powdered form of all compounds were analyzed in the IR spectrum to interpret the presence of different functional groups, using the Perkin Elmer FT-IR spectrometer (Boston, MA, USA), in the absorbance mode.

### 4.7. Atomic Force Microscopy (AFM)

A total of 5 μL of the samples (1 mM) were deposited onto a freshly cleaved muscovite Ruby mica sheet (ASTM V1 Grade Ruby Mica from MICAFAB, Chennai, India) for 5–10 min and then the sample was dried by using a vacuum dryer. An AAC mode AFM was performed using a Pico plus 5500 AFM (Agilent Technologies Santa Clara, CA, USA), using a piezo scanner, with a maximum range of 9 μm. Microfabricated silicon cantilevers of 225 μm in length with a nominal spring force constant of 21–98 N m^−1^ from nanosensors were used. The cantilever oscillation frequency was tuned into resonance frequency. The cantilever resonance frequency was 150–300 kHz. The images (256 by 256 pixels) were captured with a scan size between 0.5 and 5 μm at a scan rate of 0.5 lines per s. The images were processed by flatten using Pico view1.4 version software (Agilent Technologies, Santa Clara, CA, USA). Image analysis was done through Pico Image Advanced version software (Agilent Technologies, Santa Clara, CA, USA).

### 4.8. Transmission Electron Microscopy (TEM)

A freshly prepared solution of the M-Vit.E-CD-CS NPs in double-distilled water was placed on a TEM grid (300-mesh carbon-coated Cu grid). The samples were allowed to dry in air at room temperature for a few hours before the measurements were recorded.

### 4.9. Differential Scanning Calorimetry (DSC) and X-ray Powder Diffraction (XRD)

The thermal and crystallographic characterization was performed by DSC and XRD, respectively. Thermogravimetric analysis (TGA) was performed using a TGA Q500 system from TA Instruments Inc. (New Castle, DE, USA) under an N2 atmosphere from 0–500 °C at a heating rate of 5 °C/min. Differential Scanning Calorimetry (DSC) was performed using a DSC Q200 RCS system from TA Instruments Inc., with a refrigerated cooling system. The sample was heated with a constant ramp rate of 10 °C/min between −30 °C and 90 °C.

### 4.10. Nuclear Magnetic Resonance (NMR) Spectroscopy

A total of 2 mg of MOR, Vit.E, β-CD, and chitosan were dissolved using different solvents according to their solubility. Then, 1H NMR was recorded.

### 4.11. In Vitro Drug Release Studies

M-Vit.E-CD-CS NPs was solubilized in a phosphate-buffered saline (PBS) medium (1 M, NaCl = 8 gm, KCl = 0.2 gm, Na_2_HPO_4_ = 1.44 gm, KH_2_PO_4_ = 0.24 gm were dissolved in double-distilled water and the pH was adjusted to 7.4). To study the in vitro release kinetics, M-Vit.E-CD-CS NPs (10 mg) was filled in the dialysis bag of a cut off size of 5 kDa, put in a 200 mL of PBS of pH 7.4, and stirred at 1 g for 4 days. A total of 2 mL of the buffer solution was removed after a fixed time interval and then replaced with a fresh buffer. Using a UV-visible spectrophotometer (JASCO V-730 spectrophotometer, Tokyo, Japan) at 270 and 395 nm, respectively, the release of drugs were tested. The experiments were repeated thrice, and the average values were evaluated.

### 4.12. Cell Culture

HepG2 (human hepatocellular carcinoma) cells were grown in DMEM with 10% fetal bovine serum and 1% pen-strep at 37 °C at 5% CO_2_ in a humid environment. After 60–70% confluency, cells were treated with arsenic for 24 h and cotreated without and/or with MOR, vitamin E, and M-Vit.E-CD-CS NPs. Then, an MTT assay was done for cell survivability and ROS was measured by a fluorometry assay with DCFDA. Cell imaging study was done using nuclear staining DAPI.

### 4.13. Animals

Male BALB/c mice of 6–8 weeks, weighing 20–22 g were procured from the animal house division of our institute (CSIR-Indian Institute of Chemical Biology, Kolkata, India) and fed with standard chow diets and drinking water. The animals were kept at 22−24 °C temperature, 50−60% humidity, and subject to light and dark cycles of 12:12 h. The research procedure conducted on animals was as per the recommendations of the CSIR-Indian Institute of Chemical Biology Animal Ethics Committee.

### 4.14. Dose Selection of MOR, Vitamin E and M-Vit.E-CD-CS NPs

We used the Morin dose from a previous study [24]; and, for the dose selection of vitamin E and M-Vit.E-CD-CS NPs, each group of mice were treated with 0, 25, 50, 100 mg/kg body weight of vitamin E or 0, 10, 20, 40 mg/kg body weight of M-Vit.E-CD-CS NPs on every alternate day up to 28 days by oral gavage, after acclimatization for 7 days. AST and ALT were measured at different time intervals. Of doses of vitamin E or M-Vit.E-CD-CS NPs, those that did not elevate the level of AST or ALT were selected for experimental study.

### 4.15. Experimental Design

The experimental plan of this study was as follows: Mice were exposed to arsenic (40 mg/L) only via drinking water for 30 days [29]. Arsenic-exposed mice were orally given 0, 50, 100, 200 mg/kg body weight of MOR; 0, 25, 50, 100 mg/kg body weight of vitamin E; and 0, 10, 20, 40 mg/kg body weight of M-Vit.E-CD-CS NPs. MOR, vitamin E, and M-Vit.E-CD-CS NPs were given on every alternate day by suspending in 0.1% Tween 80 in PBS starting from day 2 to 28 days of arsenic exposure. Blood was drawn by tail vein puncture at different time points from each group of mice, for serum analysis. The mice were sacrificed after 30 days and the livers sections were collected for histology and Western blot analysis. For biochemical testing, one part of the liver was stored at −80 °C freezer and the other part was cut and placed in a bottle containing 10% neutral-buffered saline for performing histopathological analysis the next day.

### 4.16. Measurement of Serum ALT, AST, HDL, LDL, TG, TC, TC, TG, LDL, HDL, Uric Acid, Creatinine, and MDA

Blood samples were obtained by tail-vein puncture and kept at 4 °C undisturbed o/n. Samples were centrifuged the next day (1100× *g*, 10 min, 4 °C) to obtain the serum from the experimental groups. Serum AST, ALT, ALP, HDL, LDL, TG, TC, SOD, GSH, and catalase activities were calculated according to the instruction brochure provided with the commercial assay kits.

### 4.17. Assessment of Arsenic Deposition in Organs

Levels of arsenic in the tissue samples of As-exposed mice were measured by the method previously described [38]. Briefly, liver and spleen samples were taken in a 15 mL polypropylene tube in the presence of 3 mL of nitric acid (61%). The tubes were capped properly and incubated at 80 °C for 48 h, followed by cooling for 1 h to room temperature. After cooling, 3 mL of hydrogen peroxide (30%) was added to each tube, followed by incubation at 80 °C for 3 h. After suitable dilution of the digested materials with ultrapure water, levels of arsenic in the samples were determined by the colorimetric method based on the reaction of As (III) with potassium iodate in an acid medium to liberate iodine. This liberated iodine bleaches the orangish red color of Rhodamine B. Sample solution (1 mL) obtained after digestion of the tissue was mixed with 2 mL of KIO_3_ and 1 mL of HCl, and the mixture was gently shaken, followed by addition of 0.1% Rhodamine-B. The solution was kept for 15 min. The absorbance was measured at 550 nm. A decrease in absorbance is directly proportional to the concentration arsenic (III).

### 4.18. Determination of ROS

Liver samples (200 mg each) were homogenized (1:10 *w/v*) in Tris-HCl buffer (40 mM, pH = 7.4, 0 °C). One hundred mililiters of tissue homogenate was mixed with 1 mL of Tris-HCl buffer and 5 μL of 2,7-dichlorofluorescein diacetate solution (10 mM). The mixture was incubated for half an hour at 37 °C. Finally, the sample fluorescence intensity was measured using a spectrofluorometer at 480 and 525 nm wavelengths of excitation and emission.

### 4.19. Estimation of Lipid Peroxidation Levels in the Liver

The level of lipid peroxidation in liver tissues was measured as the amount of thiobarbituric acid reactive substances (TBARS) [39]. Thiobarbituric acid reacts with MDA (malondialdehyde), a major lipid oxidation product to form a red product (TBARS) that can be detected colorimetrically at 532 nm or fluorometrically at Ex/Em 532/553 nm. Briefly, supernatants of tissue lysate were mixed with an equal volume of TCA-BHT (BHT = butylated hydroxytoluene) in order to discard proteins. BHT stops further sample peroxidation during the experimental process. After centrifugation (1000× *g*, 10 min, 4 °C), 200 mL of the resulting supernatant was mixed with 40 mL of HCl (0.6 M) and 160 mL of thiobarbituric acid (TBA) 20% dissolved in Tris. The mixture was heated at 80 °C for 10 min and after cooling at room temperature the absorbance was read at 530 nm and TBARS values were calculated and expressed in nmol/mg protein.

### 4.20. Estimation of SOD, GSH, Catalase

Blood samples were obtained by tail-vein puncture and kept at 4 °C undisturbed o/n. Samples were centrifuged the next day (1100× *g*, 10 min, 4 °C) to obtain the serum from the experimental groups. Serum, SOD, GSH, and catalase activities were calculated according to the instruction brochure provided with the commercial assay kits.

### 4.21. Histological Evaluation

The fixed liver tissues in 10% neutral buffered formalin (NBF) was embedded in paraffin, thinly sectioned, de-paraffinated, and rehydrated using the standard histology procedure. Various pathological changes were assessed by using hematoxylin and eosin stains. The damage scores were estimated by counting the morphological alterations in 10 randomly selected microscopic fields from 6 samples of each group and from at least 3 independent experiments. The morphological liver integrity was graded on a scale of 1 (excellent) to 5 (poor). Liver damage scores were adopted from the study of t’Hart et al. [27] and described as: (1) normal rectangular structure, (2) rounded hepatocytes with an increase of the sinusoidal spaces, (3) vacuolization, (4) nuclear picnosis, and (5) necrosis.

### 4.22. Assessment of Serum Cytokines

Blood samples were isolated at different time points as mentioned above, and the serum levels of TNF-α, IL-1β, TGF-β, and IL-6 were determined, using the commercially available ELISA kit according to the manufacturer’s instruction and guidelines (R&D Systems, Minneapolis, MN, USA).

### 4.23. Tissue Distribution Study

All mice were fasted overnight and fed only water before the experiments. Standard stock solutions of M-Vit.E-CD-CS NPs (1 mg/mL) were prepared by dissolving the specific amounts of the drug in ethanol. After oral administration of M-Vit.E-CD-CS NPs, mice were sacrificed at 2, 6, 12, 24, 48, and 72 h. Various tissues (liver, lung, kidney, spleen) were collected and washed with 0.9% NaCl to remove the extra blood and contents. After blotting them with filter paper, 1 mg equivalent from the tissues was weighed and homogenized in 1 mL of 0.9% NaCl. Then, 100 μL of it was used as the tissue sample. Blood samples were drawn from the tail vein and coagulated for half an hour in an MCT tube. The blood samples were centrifuged at 2000 rpm for 10 min (4 °C), and serum was obtained from the supernatant. Then, 100 μL of the serum was used as the sample. Tissues were stored at −80 °C for further use [40].

### 4.24. Western Blot Analysis

Dissected tissues frozen in liquid nitrogen were disrupted using a homogenizer and RIPA lysis buffer and then centrifuged. Protein concentrations in the supernatant of tissue lysate were assessed using the Bradford method. Western blot analysis was done using 25 μg of total protein from a tissue homogenate. SDS-PAGE was carried out on an acrylamide gel to separate the proteins, which were then transferred to a polyvinylidene difluoride membrane. The membrane is blocked with 10% skimmed milk or 5% BSA and incubated overnight with primary antibody of different proteins and β-actin (1:2000 dilution; Santa Cruz, CA, USA), and next day after washing the membrane 3 times with wash buffer, was incubated with the alkaline phosphatase conjugated secondary antibody (1:5000 dilution) for 2 h. At last, protein expressions were detected using an NBT/BCIP solution.

### 4.25. Statistical Analysis

All data from at least three experiments with replicates were expressed as mean standard deviation (SD). Using GraphPad Prism software (San Diego, CA, USA), statistical significance and differences between the control and five other treatment groups were analyzed using a one-way analysis of variance.

## 5. Conclusions

In this study, as a new carrier, M-Vit.E-CD inclusion complex loaded chitosan nanoparticles were designed and synthesized, which improved the solubility of MOR and vitamin E. Using readily available Morin and vitamin E as a hydrophobic drug and β-CD, chitosan as starting materials, the MOR-Vit.E-β-CD inclusion complex loaded chitosan nanoparticles were effectively synthesized by an ionic gelation method. In vitro release study indicates that MOR and vitamin E are released pH-dependently from the nanoparticles in a sustainable way. Therefore, the newly synthesized drug carrier has a promising effect as a biodegradable transport system for sustained release of hydrophobic drugs in a pH-dependent manner. Our study indicates that M-Vit.E-CD-CS NPs have a better hepatoprotective effect than free MOR and vitamin E against arsenic-intoxicated liver injury in a murine model. The better protective effect of M-Vit.E-CD-CS NPs can be attributed to the improved solubility and bioavailability of MOR and vitamin E due to the formation of a nanocarrier. The antioxidant, anti-inflammatory, and anti-apoptotic effects of M-Vit.E-CD-CS NPs can be accounted as primary reasons for the hepatoprotective activity. Therefore, MOR-Vit.E-β-CD inclusion complex loaded CS nanoparticles can be used as a potential therapeutic agent for liver damage due to arsenic intoxification.

## Figures and Tables

**Figure 1 molecules-27-05819-f001:**
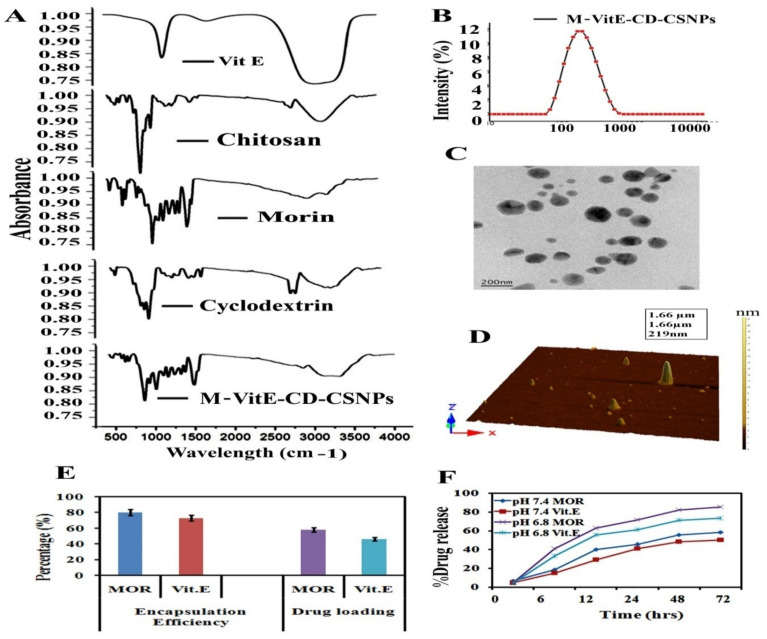
Characterization of Morin, vitamin E, and β-CD inclusion complex loaded chitosan nanoparticles. (**A**) Fourier transform infrared spectroscopy (FTIR) spectra of Morin, vitamin E, β-CD, chitosan, and M-Vit.E-CD-CS NPs. (**B**) Particle size distribution from differential light scattering (DLS) with M-Vit.E-CD-CS NPs. (**C**) TEM images of M-Vit.E-CD-CS NPs. (**D**) M-Vit.E-CD-CS NPs particle surface topology determination using atomic force microscopy (AFM). The acquired images were analyzed using scanning probe microscopy (SPM) tools for laboratory study. (**E**) The encapsulation efficiency percentage and the drug-loading percentage of M-Vit.E-CD-CS NPs. (**F**) The percentage of release of Morin from M-Vit.E-CD-CS NPs over a time period of 0–72 h. Result is the mean ± standard deviation (SD) from triplicate independent experiments.

**Figure 2 molecules-27-05819-f002:**
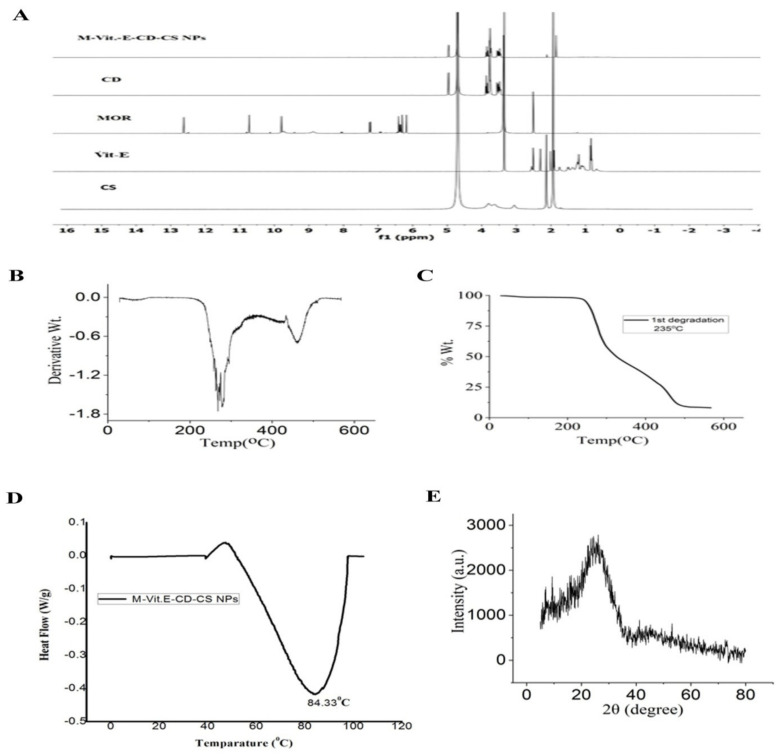
(**A**) 1HNMR spectra of MOR, Vitamin E, β-CD, chitosan, and M-Vit.E-CD-CS NPs. (**B**,**C**) Thermogravimetric analysis (TGA) was performed using a TGA Q500 system from TA Instruments Inc. under a N2 atmosphere from 0–500 °C at a heating rate of 5 °C/min. (**D**) Differential Scanning Calorimetry (DSC) was performed using a DSC Q200 RCS system from TA Instruments Inc. (New Castle, DE, USA), with a refrigerated cooling system. The sample was heated with a constant ramp rate of 10 °C/min between −30 °C and 90 °C. (**E**) XRD: X-Ray Diffraction.

**Figure 3 molecules-27-05819-f003:**
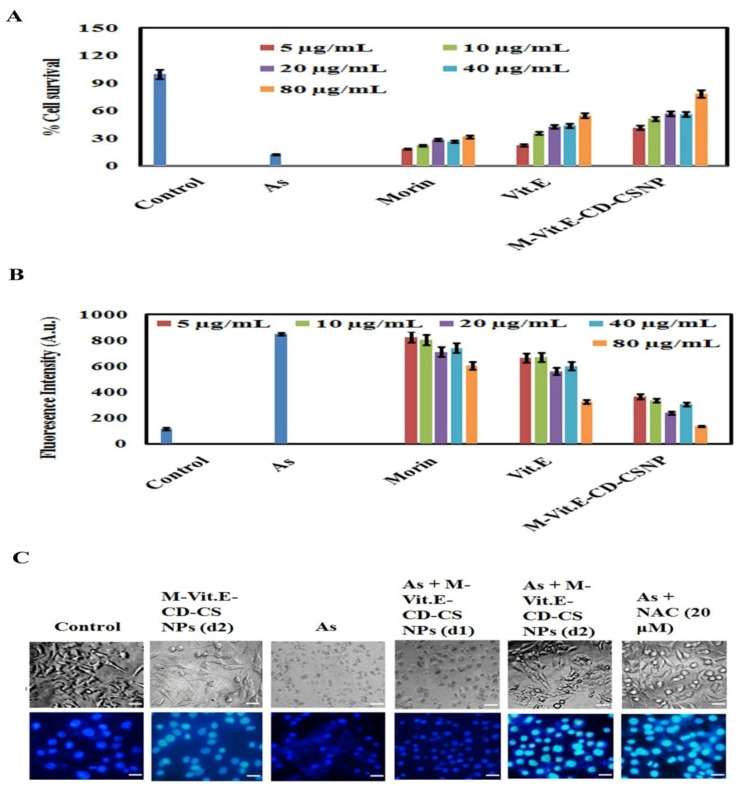
Effect of MOR, vitamin E, and M-Vit.E-CD-CS NPs on arsenic treated HepG2 cells. (**A**) % cell survivability was measured by MTT assay. (**B**) ROS level was detected fluorometrically. (**C**) Cell morphology was analyzed, using nuclear staining DAPI and bright field (Scale bar 10 μM).

**Figure 4 molecules-27-05819-f004:**
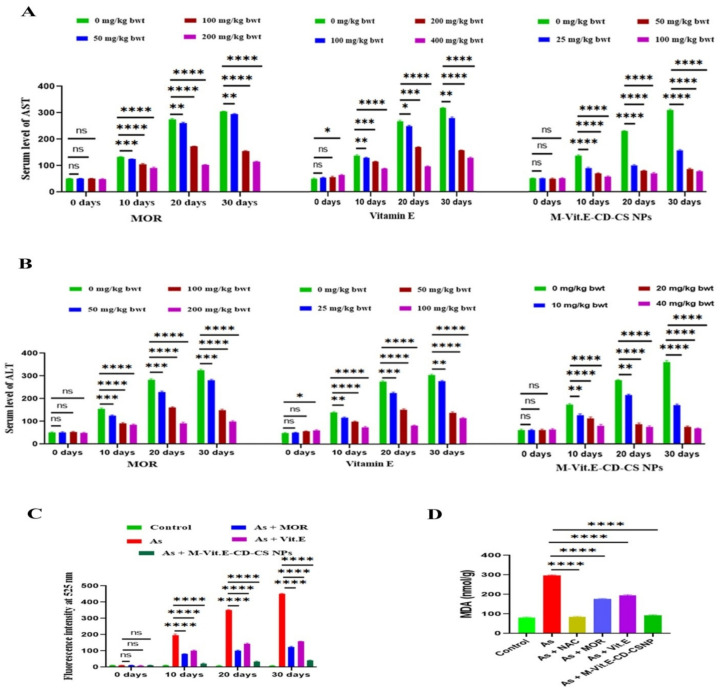
Effect of MOR, vitamin E, and M-Vit.E-CD-CS NPs on arsenic-induced elevation of (**A**) ALT, (**B**) AST, (**C**) ROS generation, and (**D**) MDA level. MOR, vitamin E, and M-Vit.E-CD-CS NPs level with the duration of its exposure. Indicated doses of MOR, vitamin E, and M-Vit.E-CD-CS NPs were treated during arsenic exposure. Data is one of the three representative experiments ± SD (ns—nonsignificant,* *p*, 0.1, ** *p*, 0.01, *** *p*, 0.001, **** *p*, 0.0001).

**Figure 5 molecules-27-05819-f005:**
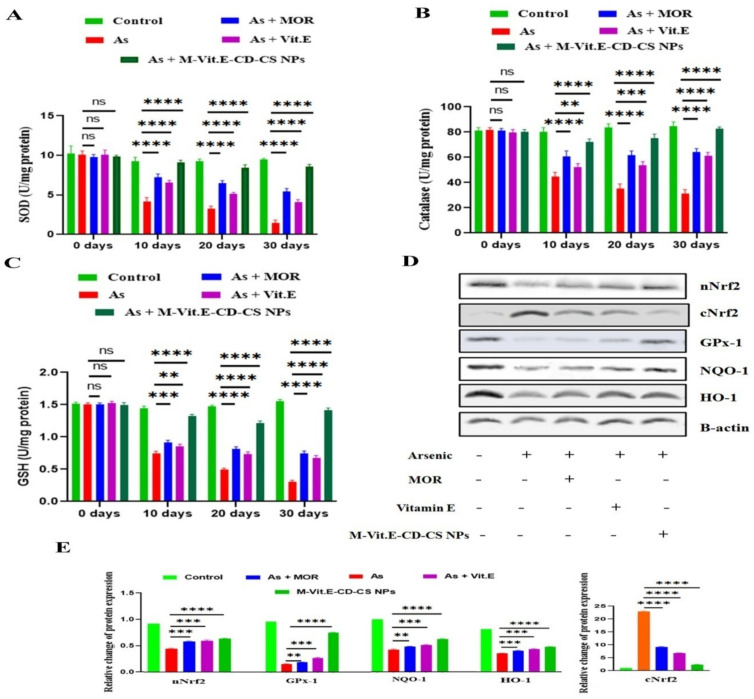
Effect of MOR, vitamin E, and M-Vit.E-CD-CS NPs on antioxidant factors. M-Vit.E-CD-CS NPs (20mg/kg bwt), vitamin E (50 mg/kg bwt), and MOR (200 mg/kg bwt) were orally treated in mice during their exposure to arsenic. The level of (**A**) SOD (**B**) catalase (**C**) GSH in the liver tissue lysate of arsenic was measured by using assay kits (ns—nonsignificant, ** *p*, 0.01, *** *p*, 0.001, **** *p*, 0.0001). (**D**) the effect of MOR, vitamin E, and M-Vit.E-CD-CS NPs on protein expression of cytosolic Nrf2, nuclear Nrf2, GPx, HO-1, and NQO1 (western blot analysis) in the liver tissue lysate. (**E**) Relative change of protein expression of the respective protein with respect to β-actin (** *p*, 0.01, *** *p*, 0.001, **** *p*, 0.0001).

**Figure 6 molecules-27-05819-f006:**
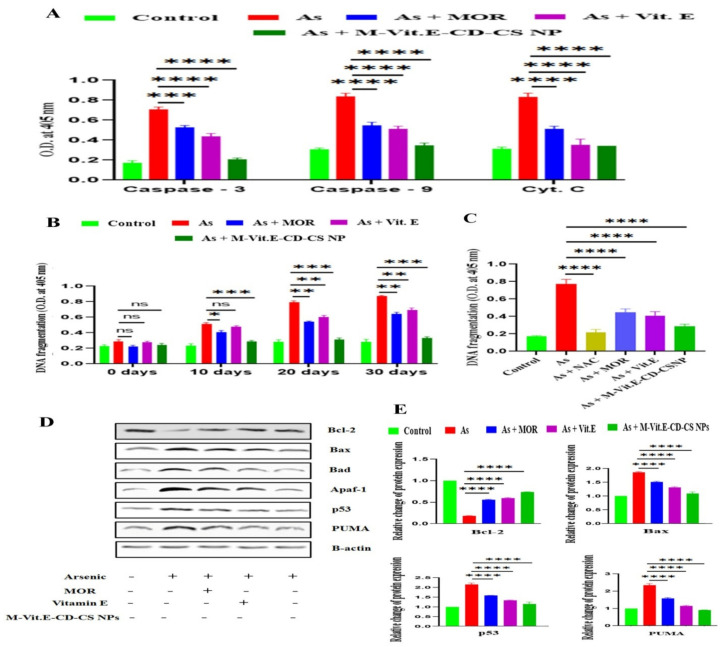
Effect of MOR, vitamin E, and M-Vit.E-CD-CS NPs on liver tissue apoptosis. (**A**) The level of active caspase-3, active caspase-9, and cytosolic cytochrome C obtained using respective colorimetric assay kits (*** *p*, 0.001, **** *p*, 0.0001). (**B**,**C**) Level of DNA fragmentation obtained using DNA fragmentation kit (ns—nonsignificant, * *p*, 0.1, ** *p*, 0.01, *** *p*, 0.001, **** *p*, 0.0001). (**D**) the effect of MOR, vitamin E, and M-Vit.E-CD-CS NPs on protein expression (western blot analysis) of Bcl-2, Bax, Bad, p53, Apaf-1, and PUMA in mice exposed to arsenic. (**E**) Relative change of protein expression of the respective protein with respect to β-actin (**** *p*, 0.0001).

**Figure 7 molecules-27-05819-f007:**
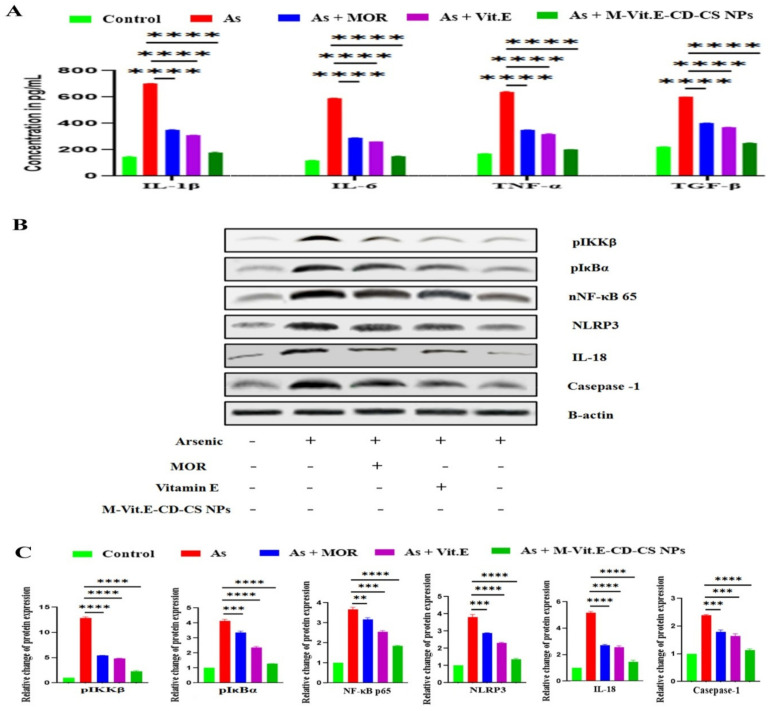
Effect of MOR, vitamin E, and M-Vit.E-CD-CS NPs on the level of (**A**) TNF-α, IL-β, IL-6, and TGF-β in the liver tissue lysate of arsenic challenged mice as seen in ELISA analysis. Data are one of the three representative experiments ± SD (**** *p*, 0.0001). (**B**) the effect of MOR, vitamin E, and M-Vit.E-CD-CS NPs on protein expression of nuclear NF-kBp65, NLRP3, Caspase-1, and IL-18 (western blot analysis). (**C**) Relative change of protein expression of the respective protein with respect to β-actin (** *p*, 0.01, *** *p*, 0.001, **** *p*, 0.0001).

**Figure 8 molecules-27-05819-f008:**
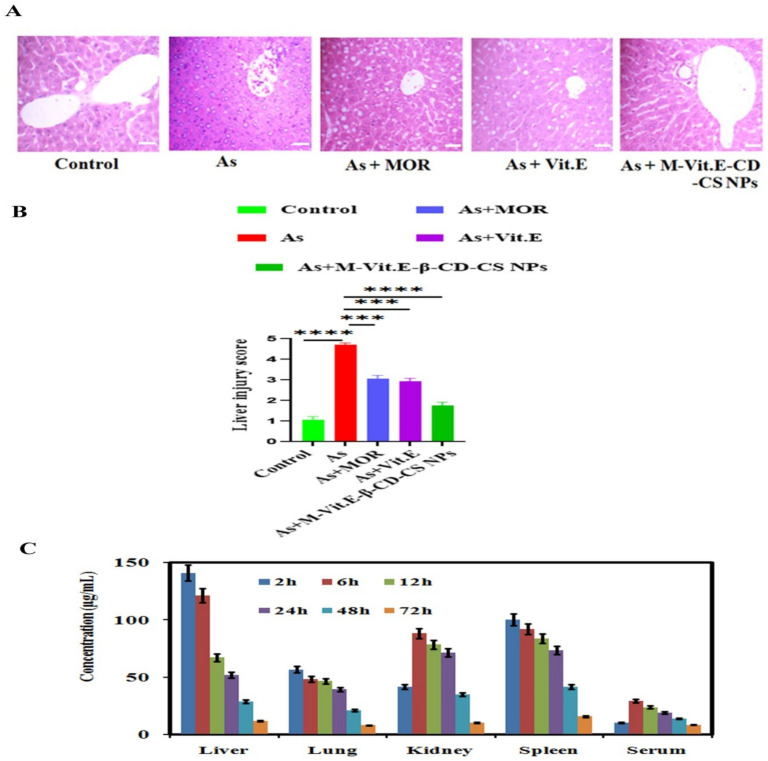
(**A**) Effect of MOR, vitamin E, and M-Vit.E-CD-CS NPs on liver tissue histology architecture of liver tissue section, following treatment without or with MCNPs and MOR in arsenic-exposed mice (Scale bar 100 microns). (**B**) Histopathological scoring of the liver tissue (*** *p*, 0.001, **** *p*, 0.0001). (**C**) Tissue distribution studies of M-Vit.E-CD-CS NPs in various organs. Mean concentration of Morin in liver, lungs, kidneys, spleen, and serum at 2, 6, 12, 24, 48, and 72 h after oral administration of single dose of M-Vit.E-CD-CS NPs (*n* = 3, mean ± SD) in mice.

**Table 1 molecules-27-05819-t001:** Effect of oral administration of MOR and MCNPs on hematological parameters.

Parameters	Control Group	Arsenic (40 mg/L) Treated Mice	Arsenic + MOR (200 mg/kg) Treated Mice	Arsenic + Vitamin E (50 mg/kg) Treated Mice	Arsenic + M-Vit.E-CD-CS NPs (20 mg/kg)
Body weight gain (gm)	0.52 ± 0.06	0.29 ± 0.03	0.45 ± 0.04	0.39 ± 0.02	0.49 ± 0.03
RBC: No. of cells (10^6^/μL)	8.1 ± 0.53	5.2 ± 0.35	6.8 ± 0.27	6.1 ± 0.33	7.5 ± 0.41
WBC: No. of cells (10^3^/μL)	12.6 ± 0.13	16.2 ± 0.31	15.1 ± 0.24	15.8 ± 0.41	13.1 ± 0.22
Hb (gm/dl)	13.6 ± 0.38	11.7 ± 0.23	12.5 ± 0.47	12.1 ± 0.17	12.9 ± 0.38
PLT (10^3^/μL)	556 ± 31.2	412 ± 28.2	512 ± 25.8	516 ± 36.2	545 ± 19.7
LDH (U/L)	401 ± 24.3	742 ± 31.3	495 ± 14.5	481 ± 16.6	412 ± 11.3
Uric acid (mg/dL)	2.61 ± 0.72	4.73 ± 0.51	3.09 ± 0.34	3.21 ± 0.51	2.76 ± 0.42
Creatinine (mg/dL)	0.51 ± 0.04	2.8 ± 0.13	1.7 ± 0.09	1.5 ± 0.06	0.61 ± 0.03
Cholesterol (mg/dL)	145 ± 7.6	281 ± 31.2	162 ± 11.2	173 ± 16.3	151 ± 4.5
TG (mg/dL)	85.1 ± 8.2	163 ± 12.1	92.4 ± 9.1	104.4 ± 6.2	89.3 ± 3.8
HDL (mg/dL)	61.8 ± 2.9	35.6 ± 4.1	45.1 ± 2.3	48.4 ± 3.1	55.3 ± 3.5
LDL (mg/dL)	76.2 ± 5.7	165.6 ± 10.8	92.5 ± 6.1	95.6 ± 5.3	81.1 ± 4.6
Phospholipid (mg/dL)	45.5 ± 7.5	22.1 ± 4.3	38.1 ± 2.9	35.3 ± 3.1	42.5 ± 3.1

Values are expressed as mean ± SEM (*n* = 3). *p* > 0.05 when compared to normal group.

**Table 2 molecules-27-05819-t002:** Arsenic deposition in different organ.

Arsenic Concentration in μg/g of Tissue in 30 Days						
	Liver	Kidney	Cerebellum	Lung	Heart	Skin
Arsenic, 40 mg/L	145.3 ± 5.1	29.8 ± 1.8	12.5± 4.4	14.6 ± 1.3	12.8 ± 5.6	4.8 ± 2.6
Arsenic + MOR (200 mg/kg)	110.4 ± 2.5	18.1 ± 1.4	8.2 ± 2.1	9.9 ± 1.9	8.5 ± 2.2	1.9 ± 0.6
Arsenic + vitamin E (50 mg/kg)	95.2 ± 2.4	15.1 ± 1.6	6.6 ± 1.3	7.1 ± 1.1	6.1 ± 0.8	0.7 ± 0.16
Arsenic + M-Vit.E-CD-CS NPs (20 mg/kg)	41.6 ± 3.3	3.1 ± 1.2	0.52 ± 0.07	1.2 ± 0.03	1.5 ± 0.21	0

Values are expressed as mean ± SEM (*n* = 3). *p* > 0.05 when compared to normal group.

## Data Availability

Data is contained within the article.

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
