# Peer review of "Morin-VitaminE-β-CyclodextrinInclusionComplexLoadedChitosanNanoparticles (M-Vit.E-CD-CSNPs) Ameliorate Arsenic-Induced Hepatotoxicityina Murine Model"

_molecules, 2022, doi:10.3390/molecules27185819_

Round 1

Reviewer 1 Report

This manuscript by Saha and colleagues designed a new nano-carrier achieved by solubility of MOR/Vit-E such as MOR-Vitamin E-β-CD inclusion complex loaded chitosan nanoparticles and evaluated its hepatoprotective effect by inhibition of inflammation, apoptotic responses as well as antioxidant enhancement. As well as the authors claim that this system display better protective than free MOR/Vit-E against arsenic-induced hepatotoxicity in mice. Although the work is an interesting piece of research and overall amount of novelty is not remarkably high, however, in my opinion, ample to allow publication, after addressing a
few issues.

Concerns to authors:

1.     I suggest authors to include better title and abstract must be rephrase it, main findings should be focused, and others moved to introduction and reference must be removed from abstract.

2.     This referee doesn’t know much idea about MOR-Vitamin E-β-CD and strongly encourage to include pictorial representations of those as a figure with respect to readers.

3.     Authors should perform experiments for inclusion phenomenon  (NMR, DSC and XRD).

4.      Visual presentation of the water-solubility has to be included.

5.     In Fig 2 , A & B included with better resolution, Line 116—should be results 2, not 1, line 454-- ml—mL , minor typos should be thoroughly rechecked.

Author Response

Reviewer 1:

  1. I suggest authors to include better title and abstract must be rephrase it, main findings should be focused, and others moved to introduction and reference must be removed from abstract.

Answer: Thank you for your valuable comments. We have changed the title of the manuscript. References are removed from the abstract. The abstract has rephrased again with focusing the main findings.

  1. This referee doesn’t know much idea about MOR-Vitamin E-β-CD and strongly encourage including pictorial representations of those as a figure with respect to readers?

Answer: A graphical abstract is attached for the better understanding of our work.

  1. Authors should perform experiments for inclusion phenomenon (NMR, DSC and XRD).

Answer: NMR, XRD and DSC data are attached as Figure 2.

  1. Visual presentation of the water-solubility has to be included.

Answer: A video of the preparation of the nanoparticles in two steps are attached.

  1. In Fig 2 , A & B included with better resolution, Line 116—should be results 2, not 1, line 454-- ml—mL , minor typos should be thoroughly rechecked.

Answer: Better resolution of Figure 3A & B (previously marked as Figure 2A & B) is included. We have checked the draft of the paper thoroughly to omit the minor typos error.

Reviewer 2 Report

The article ”Hepatoprotective effect of Morin-Vitamin E-β-Cyclodextrin inclusion complex loaded Chitosan nanoparticles (M-Vit.E-CD-CS NPs) through improvement of antioxidant system, prevention of apoptosis and inflammation in arsenic-induced liver injury in mice” has an appropriate subject and it is suitable for the purpose and objective of Molecules.

The research plan is thoroughly designed and methodically described, and there is plenty of data presented.

In order to be published I have some suggestions for the authors:

-        the title could be shorter, like: ”Hepatoprotective effect of Morin-Vitamin E-β-Cyclodextrin inclusion complex loaded Chitosan nanoparticles in arsenic-induced liver injury in mice”

-        simplify the introduction, it is too long

-        figure 1: the FTIR spectra is in the mirror (usually it is from 4000 to 400 cm-1)

-        pay attention at all the editing: sections 2.3 to 2.6 are not in the appropriate style; there are sentences with a break between the words (example: lines 309-310; 420-421, and others)

-        keep the discussion sections about your work, without repeating ideas from introduction

Reviewer 3 Report

The manuscript “Hepatoprotective effect of Morin-Vitamin E-β-Cyclodextrin inclusion complex loaded Chitosan nanoparticles (M-Vit.E-CD-CS NPs) through improvement of antioxidant system, prevention of apoptosis and inflammation in arsenic-induced liver injury in mice” by Sanchaita Mondal et al is very interesting. They have prepared Morin vitamin E-β-Cyclodextrin inclusion complex loaded Chitosan nanoparticles (M-Vit.E-CD-CS NPs). They have reported that prepared NPs could be a therapeutic agent that can protect liver against arsenic toxicity. To prove their hypothesis, they have performed FTIR, biochemical assays, ELISA western blot and histological analysis. I have several major comments that needs to be addressed.

Comments:

1.      The manuscript is written in poor English,

2.      Authors have put the references in abstract section, although abstract must be self-contained and comprehensible without the need of outside sources or references to the actual document. So, authors must remove references from the abstract.   

3.      Introduction section line 92 and 93 are not consistent, line 105 to 110 have not been cited and in line 99, authors have described GSH as an enzyme. Is GSH enzyme???

4.      Magnification should be added in Figure 2C.

5.      How the doses of arsenic and NPs were decided?

6.      Authors have not applied statics anywhere throughout the manuscript, which is the drawback of the manuscript.

7.      How much proteins were used and what was the dilution of secondary antibodies used for western blot analysis. Authors should add in in manuscript.

8.      Molecular weight of all proteins in missing in figure 4D,5D and 6B, must be added and quantifications of all the western blot data must be added in the manuscript.

9.      In figure 7A, the image of As group is looking different than all others, authors should add all the images of all the groups in same orientation and must add scale bar and magnifications. Also, I will suggest to add scoring system to the histopathological analysis to access the severity and protection by NPs .

10.  I will suggest adding a graphical abstract, which is the point of attraction for a reader.

Author Response

Reviewer 3:

  1. The manuscript is written in poor English

Answer: Thank you for your valuable comments. We have re-write the abstract, introduction and discussion part with better English as per our understanding.

  1. Authors have put the references in abstract section, although abstract must be self-contained and comprehensible without the need of outside sources or references to the actual document. So, authors must remove references from the abstract.   

Answer: References are removed from the abstract part.

  1. Introduction section line 92 and 93 are not consistent, line 105 to 110 have not been cited and in line 99, authors have described GSH as an enzyme. Is GSH enzyme???

Answer: Actually GSH is not an enzyme so I have changed the sentence and replaced the word “anti-oxidant enzymes” by “anti-oxidant factors”.

  1. Magnification should be added in Figure 2C.

Answer: Magnification is added in Figure 3C (previously marked as Figure 2C)

  1. How the doses of arsenic and NPs were decided?

Answer: The dose of arsenic was decided according to our previous report and it was added in reference no 25. Selection of nontoxic dose of NPs was given in the supplementary figure S2 & S3 and its explanation was given in the section 4.14.

  1. Authors have not applied statics anywhere throughout the manuscript, which is the drawback of the manuscript.

Answer: We have added the statistical analysis data in our revised manuscript.

  1. How much proteins were used and what was the dilution of secondary antibodies used for western blot analysis. Authors should add in manuscript.

Answer: We have added the total protein concentration which was used for the western blot analysis and the dilution factor of the secondary antibody in the section 4.24.

  1. Molecular weight of all proteins in missing in figure 4D, 5D and 6B, must be added and quantifications of all the western blot data must be added in the manuscript.

Answer: Molecular weights of all proteins were given in the original full blot. We have not added it in the main figure due to clumsiness. Quantifications of all western blot data were included in figures.

  1. In figure 7A, the image of As group is looking different than all others, authors should add all the images of all the groups in same orientation and must add scale bar and magnifications. Also, I will suggest to add scoring system to the histopathological analysis to access the severity and protection by NPs.

Answer: In figure 8A (previously marked as 7A), due to apoptosis the cell shrinkage was occurred and also the sinusoids line was not in ordered which is the indication of liver damage. The scale bar and scoring to the histopathological analysis was added in the main figure.

  1. I will suggest adding a graphical abstract, which is the point of attraction for a reader.

Answer: Graphical abstract is added for better understanding of our work.

Round 2

Reviewer 1 Report

The authors have included suggested changes in the revised version and happy to recommend it for acceptance. Congratulates to Dr. Saha and team for their nice work. 

.with best wishes

Reviewer 3 Report

I have no more comments